# Antimicrobial Properties of Different Hop (*Humulus lupulus*) Genotypes

**DOI:** 10.3390/plants12010120

**Published:** 2022-12-26

**Authors:** Zala Kolenc, Tomaž Langerholc, Gregor Hostnik, Miha Ocvirk, Sara Štumpf, Maša Pintarič, Iztok Jože Košir, Andreja Čerenak, Alenka Garmut, Urban Bren

**Affiliations:** 1Laboratory of Physical Chemistry and Chemical Thermodynamics, Faculty of Chemistry and Chemical Technology, University of Maribor, Smetanova Ulica 17, SI-2000 Maribor, Slovenia; 2Department of Microbiology, Biochemistry, Molecular Biology and Biotechnology, Faculty of Agriculture and Life Sciences, University of Maribor, Pivola 10, 2311 Hoče, Slovenia; 3Department for Agrochemistry and Brewing, Slovenian Institute of Hop Research and Brewing, Cesta Žalskega Tabora 2, SI-3310 Žalec, Slovenia; 4Department for Plants, Soil and the Environment, Slovenian Institute of Hop Research and Brewing, Cesta Žalskega Tabora 2, SI-3310 Žalec, Slovenia; 5Department of Applied Natural Sciences, Faculty of Mathematics, Natural Sciences and Information Technologies, University of Primorska, Glagoljaška Ulica 8, SI-6000 Koper, Slovenia; 6Institute of Environmental Protection and Sensors, Beloruska Ulica 7, SI-2000 Maribor, Slovenia

**Keywords:** hop extract, xanthohumol, *Staphylococcus aureus*, *Lactobacillus acidophilus*, minimum inhibitory concentration (MIC), minimum bactericidal concentration (MBC)

## Abstract

The antimicrobial activity of hop extracts obtained from different hop genotypes were investigated against *Staphylococcus aureus* and *Lactobacillus acidophilus*. In this study the pure xanthohumol, purified β-acids rich fraction, as well as α-acids with β-acids rich fraction were used to test antimicrobial activity against *Staphylococcus aureus* and *Lactobacillus acidophilus*; whereby, the antimicrobial activity of different hop extracts against *Lactobacillus acidophilus* was studied for the first time. Microbial susceptibility to purified hydroacetonic extracts from different hop varieties was investigated by the broth microdilution assay to determine the minimum inhibitory concentration (MIC) and the minimum bactericidal concentration (MBC). The hop hydroacetonic extracts were more effective against *Staphylococcus aureus* than against *Lactobacillus acidophilus*. Strong inverse correlations of MIC and MBC values were obtained with xanthohumol, cohumulone, n+adhumulone, colupulone and n+adlupulone contents, suggesting that the identified chemical hop compounds are directly responsible for antimicrobial effects. Moreover, the effect of the growth medium strength on the MIC values of hop extracts against *Staphylococcus aureus* was systematically investigated for the first time. The current study also reveals the effect of different hop extracts on *Staphylococcus aureus*, which responds to their presence by lag phase extension and generation time prolongation.

## 1. Introduction

Several groups of chemical compounds originating from plants can be found in nature. These natural compounds exert a wide range of beneficial health effects [1]. Many properties of individual natural compounds are already known; nevertheless, we cannot fully exploit them until we understand their molecular mechanisms of action. On the other hand, there remain many undisclosed natural active ingredients waiting for researchers to reveal them. Likewise, bacterial infections have, in recent years, become increasingly difficult to treat due to antibacterial resistance [2,3]. Antibiotic-resistant bacteria are spreading very fast all over the world, causing many problems in the treatment of infectious diseases. Due to a non-therapeutic use, an overuse, an underuse and a general misuse of antibiotics, antibacterial resistance has evolved [4]. Action plans (as well as their implementation), in the form of restrictive measures on the use of antibiotics for human and animal health, have already taken place all around the world (World Health Organization, 2020), but this is far from solving the global challenge of antimicrobial resistance. Therefore, the discovery of new antibacterial (and antimicrobial) compounds remains crucial. Several natural compounds that possess a broad structural diversity have been reported in the scientific literature as potential antimicrobial agents or resistance-modifying agents [5]. Such phytochemical products would provide a valuable intermediate solution until new antibiotics are developed [6].

Since hop exert antimicrobial effects, the stronger hopped Indian Pale Ale style beers have been able to “survive” the shipping route from England to the English colonies, such as India [7,8]. Gram-positive bacteria (*Lactobacillus*, *Streptococcus*, *Staphylococcus*, *Micrococcus*, *Bacillus* and *Pediococcus*) can still spoil beer, increase its turbidity, and produce distasteful aromatic compounds (etc. diacetyl or/and hydrogen sulfide). Hop forms an important beer ingredient due to its potent antimicrobial effect against these spoilage bacteria [9]. Hop (*Humulus lupulus* L.) represents an industrial plant whose female inflorescences (named cones) are widely used in the brewing industry. Hop cones (Figure 1) contain various compounds such as hop resins (composed of several bitter acids), essential oils and flavonoids, which support the brewing process. Hop is used in brewing for its bitterness, flavor aroma and several additional properties (such as maintaining the microbiological stability) [10]. These properties have been recently mainly attributed to the bitter acids as well. Hop was traditionally also used for medical purposes, especially for treatment of sleeping disorders (for this purpose, hop is still used nowadays), for activation of gastric functions and as an antibacterial and antifungal agent [10]. Namely, hops contain bitter acids, which represent a great portion of dry hop cones and are further divided into α-acids (humulones) and β-acids (lupulones) [10]. In addition, flavonoids (with the main representative xanthohumol (XH)) also form important hop compounds [11] that exhibit antimicrobial activities [10,12,13] and anticarcinogenic effects [14,15].

Hop, hop extracts and individual hop components exhibited antimicrobial activities in several studies, including ours. Regarding their effectiveness, they are not as strong as antibiotics [10]. However, lupulone and XH yield a high synergistic effect with antibiotics (polymyxin B sulfate, tobramycin and ciprofloxacin) [16]. The minimal inhibitory concentration (MIC) values using the lupulone/XH combinations with antibiotics were measured, and they were significantly lower compared to pure antibiotics. These results were observed for both Gram-positive and Gram-negative bacteria. Moreover, these facts suggest the suitability of applying hops in medicine, pharmacy and veterinary medicine [10]. For nutritional purposes, hop extracts have been already applied successfully, showing an antimicrobial effect (against *Listeria monocytogenes*) in milk and some other dairy products (cottage cheese, cheese) [17]. These hop extracts also contained a high β-acids concentration. In recent research, it was revealed that hop extracts have exhibited significant activity also against acne-causing bacteria. Namely, hop CO_2_ extracts (with a high humulone and lupulone content) showed an inhibitory effect on *Propionibacterium acnes* and *Staphylococcus aureus* (one of the main acne causative bacteria) with low MIC values (3.1 and 9.4 µg/mL) [18]. The scientific literature states that feeding chickens (broilers) with hop significantly improves their growth due to its antimicrobial properties as well. Consequently, a lesser need for antibiotics to improve animal growth has been envisaged [19]. In the sugar industry, the use of β-acids from hops is recognized, as they are extremely effective in inhibiting NO_2_ formation and the development of anaerobic bacteria that spoil the products [20]. Furthermore, it was found that the hop β-acids are very useful in preventing the development of bacteria in thick juices. Although they did not exhibit a bactericidal effect, they successfully prevented the development of new bacteria in the juices [21]. Such an application of hop β-acids is very worthwhile, as they possess a less bitter taste compared to hop α-acids, which is undesirable in the food industry. Humulinic acid (an additional component of hop extracts) has no bitter taste as well, which has a very positive effect on its use for food preservation purposes. In addition, the lowest MIC value of 1.4 µM against *L*. *brevis* of all hop components investigated so far has been determined exactly for humulinic acid (derived from iso-α-acids) [10].

Several experiments in which authors determined the antimicrobial activity of various types of hop extracts have been published [7,11,18,22,23,24,25,26,27,28,29,30]. On the other hand, there are limited data on comparative studies of different hop varieties and of their impact on antimicrobial activity [25]. The methodology to reveal the antimicrobial activity of natural extracts is not stringently defined (which makes comparison of different studies difficult), in contrast to the determination of the antimicrobial activity of antibiotics [31,32,33].

*Lactobacillus acidophilus* are a Gram-positive lactic acid bacteria (LAB). *Lactobacillus* bacteria may occur in various environments, such as different food products (especially in fermented food products) as well as in the human gastrointestinal tract [34]. Selected strains of *Lactobacillus* bacteria have also been characterized as probiotic, but on the other hand they could also act as food spoilage in certain food products. It was reported that approximately 70% of all beer-spoilage is caused by LAB [9]. This happens as a consequence of adaptation of LAB to hops and hop compounds [35]. In fact, *Lactobacillus* spp. exhibited a high resistance against antibiotics (chloramphenicol, erythromycin, tetracycline, kanamycin) and, likewise, the *Lactobacillus acidophilus* showed a 42.5% resistance to penicillin [34].

For this purpose, hydroacetonic extracts (HAE) of fourteen hop varieties were obtained from Slovenia and worldwide. Our aim was to determine the antimicrobial activity of fourteen different hop varieties—of a concentrated β-acids hop extract, of a concentrated α-acids and β-acids hop extract and of pure XH—in order to show which hop component contributes the most to its antimicrobial effect against *Staphylococcus aureus* and *Lactobacillus acidophilus*. Moreover, the effect of growth medium strength on the MIC values was evaluated for all hop extracts. Furthermore, generation times and lag time durations were evaluated in accordance with the growth medium strength. To the best of our knowledge, this is the first study to determine the antimicrobial activity of hop extracts against *Lactobacillus acidophilus*.

## 2. Results and Discussion

### 2.1. Hop Extract Preparation

Our aim was to evaluate the Slovenian hop varieties in comparison to some well-known worldwide hop varieties. According to the scientific literature [22], the hop cones display an extraordinary antimicrobial activity against *Staphylococcus aureus*, compared to hop leaves. Although the major hop extracts in similar experiments were obtained by hydroethanolic solvents [12], in our study, a hydroacetonic extraction of hop cones was performed based on the comparison of the extraction yield. The extraction procedure in our study was accomplished according to the extraction procedure of Bocquet et al. [32] with some modifications. Namely, hydroacetonic extracts yielded on average 22% higher mass values compared with hydroethanolic hop extracts. The investigated hop hydroacetonic extracts were additionally purified by liquid:liquid extraction using chloroform and water and purified hydroacetonic hop extracts were obtained (HAE). There are some studies published where the chemical composition of hop extracts was not analyzed [32], or total phenolics [22]/total flavonoids [24] were only determined. These missing data preclude the understanding of which hop compound conveys the largest portion of antimicrobial activity. On the other hand, some studies of hop antimicrobial activity already include a more detailed chemical composition of hop samples, similar to our study [18,25]. The chemical composition of pure hop cones is presented in Table 1. To work with as many well-defined samples as possible, the chemical composition of HAEs, the combination of α-acids and β-acids rich fraction (αβ-AF) and the β-acids rich fraction (β-AF) were determined (Table 2 and Table 3). The isolated XH from the HAE is also included in Table 3.

### 2.2. HPLC Determination of the Chemical Composition of Purified Hydroacetonic Hop Extracts (HAE)

To understand the impact of individual components on the antimicrobial activity, the chemical analyses of HAE extracts were performed. The amounts of individual chemical components in HAE extracts are higher (Table 2) in comparison to the chemical composition of pure hop cones (Table 1), confirming a successful extraction and the feasibility of an additional purification of extracts (in our case with chloroform). In Table 1, the chemical composition of crude hop cones is reported with the aim to evaluate the primary importance of different hop varieties. The chemical composition was determined in pure hop cones to reveal the chemical composition before the extraction. The results demonstrate that Styrian Eureka contains the highest values of cohumulone (3.16%, w/w) and n+adhumulone (13.88%, w/w), Styrian Eagle contains the highest values of colupulone (2.65%, w/w) and n+adlupulones (3.91%, w/w), while the Canada variety contains the highest value of XH (4.74%, w/w). On the contrary, the Belgium variety contains the lowest values of all components except for n+adlupulne, which is the lowest in Chocotsu.

The determined chemical composition of HAE is comparable to a similar study [25], where ethanol was used as the extraction solvent and the extraction was performed at 60 °C. Our results show that the HAE of hop variety Styrian Wolf contains the highest amount of XH (2.90%, w/w), cohumulone (14.26%, w/w) and colupulone (9.45%, w/w), the HAE of Styrian Eagle contains the highest value of n+adhumulone (47.80%, w/w) and the HAE of hop variety Chocothsu contains the highest value of n+adlupulne (13.57%, w/w). On the other hand, the HAE of hop variety Styrian Eureka contains the lowest amounts of XH, cohumulone, colupulone and n+colupulone; the HAE of hop variety Canada contains the lowest amount of n+adhumulone.

### 2.3. Purification of Hop Extracts

The XH isolation from HAE was performed by using preparative High Performance Liquid Chromatography (HPLC) and 43.29 mg of XH, with which chromatographic purity of 97.99%, was obtained. The pure XH, isolated according to our protocol, was analyzed by the HPLC method. The remaining 2.01% of the material is unidentified and does not contain α-acids nor β-acids. The remaining purified extracts were commercially obtained. The β-AF, obtained from Hopsteiner, contained higher amounts of β-acids (16.90% colupulone and 15.23% n+adlupulone) than the HAE extracts of different hop varieties. On the other hand, αβ—AF, obtained from Labor Veritas, contains high amounts of α-acids (cohumulone 12.76% and n+adhumulone 72.79%), but still a non-negligible percentage of β-acids (less than 5% w/w).

### 2.4. Minimal Inhibitory Concentration (MIC) and Minimal Bactericidal Concentration (MBC) Determination

Microbial susceptibility to HAE extracts from different hop varieties was tested by the broth microdilution assay for the determination of MIC and MBC. Distinct activities against tested microorganisms were exhibited, where *Staphylococcus aureus* was found to be more sensitive to HAE compared to *Lactobacillus acidophilus*.

Hop components can penetrate through the bacterial cell wall, due to the hydrophobic nature of compounds in hop extracts. Moreover, the interaction between the hop components and the inner membrane results in cell structure damage. Consequently, the active transport of sugars and amino acids is inhibited [25]. Therefore, the greatest antibacterial activity of hop extracts is documented against Gram-positive bacteria, especially against *Staphylococcus aureus* [18,25,32,33]. Namely, the outer membrane of Gram-negative bacteria consists of lipopolysaccharides, which act as a barrier for several molecules [25]. Additionally, the recent study [32] included 20 Gram-negative bacteria to test the antimicrobial activity of crude hydroethanolic extracts of hop cone, where extracts were not effective against any of the tested bacterial strains. Nevertheless, compared to *Corynebacterium*, *Enterococcus faecalis*, *Mycobacterium smegmatis*, *Staphylococcus warneri*, *Streptococcus agalactiae* and *Streptococcus dysgalactiae*, *Staphylococcus aureus* resulted as one of the most susceptible Gram-positive bacteria toward hop antimicrobial activity [32]. In our study, the HAE extracts showed similar results (Table 4) to the described study, and Styrian Dragon exhibited the greatest antibacterial activity against *Staphylococcus aureus* (with the average MIC value of 9.8 µg/mL and the MBC value of 15.6 µg/mL). Additionally, the second highest antibacterial activity against *Staphylococcus aureus* was observed in Styrian Wolf, Aurora and Dekorativny (with average MIC values of 15.6 µg/mL and MBC values of 31.3 µg/mL). On the other hand, our results (Table 4) clearly show that Caucasus and Canada are not appropriate hops to convey the antibacterial activity against *Staphylococcus aureus*, since the average MIC values were higher than 250.0 µg/mL. The exact value could not be determined since it was impossible to dissolve higher concentrations of the HAE extract into the growth media without observable precipitation. At the same time, higher concentrations of crude hop extracts of different Brazilian varieties were observed to inhibit (MIC) and to kill (MBC) bacteria [25]. Wild hops obtained from the Caucasus and Canada revealed a low content of all five compounds determined by HPLC (Table 2). This fact indicates that XH, cohumulone, n+adhumulone, colupulone and n+adlupulone represent compounds that may confer important and significant characteristics of the antimicrobial activity of hop extracts against *Staphylococcus aureus*. Other hop varieties contain higher amounts of all five determined hop compounds, especially the cohumulone, n+adhumulone, colupulone and n+adlupulone (Table 2) and consequently exhibit an increased antimicrobial activity with lower MIC values against *Staphylococcus aureus*. Only the Styrian Eureka, on the one hand, contains low amounts of determined hop compounds and also, on the other hand, exhibits a relatively high antimicrobial activity with an average MIC value of 19.5 µg/mL, suggesting that also other compounds which were not investigated in our study may convey the antimicrobial effect. However, for the bacteria *Staphylococcus aureus*, the hop HAE extracts with average MIC values lower than 20 µg/mL, contain significantly higher amounts of the colupulon, n+adlupulone and n+adhumulone than the hop HAE extracts with average MIC values above 20 µg/mL. On this basis, we could safely assume the investigated hop compounds contribute significantly to the main antibacterial action in the mixture of hop components.

Microbial susceptibility to HAE extracts from different hop varieties was verified by broth microdilution assays to determine MIC and MBC values against *Lactobacillus acidophilus* as well. In our study (Table 4), Aurora, Savinjski golding, Styrian Dragon and Chocotsu No.17 exhibited the greatest antibacterial activity against *Lactobacillus acidophilus* (with an average MIC value of 62.5 µg/mL and MBC value of 375 µg/mL), followed by hop variety Canada P169 S369 with an average MIC value of 62.5 µg/mL and MBC value of 750 µg/mL. On the other hand, our results (Table 4) clearly display that Belgium S367 is not an appropriate variety to convey antibacterial activity against *Lactobacillus acidophilus*, since its average MIC values were higher at 208.3 µg/mL and its MBC value was 750 µg/mL. Since this is, to the best of our knowledge, the first study on the antimicrobial activity of hop extracts against *Lactobacillus acidophilus*, we could not compare our results with the scientific literature. Likewise, we cannot find a direct correlation between the content of a single compound in the hop extract and the MIC/MBC values. However, it seems that extracts with higher α-acids and β-acids content display lower MIC/MBC values. A notable exception is Styrian Eureka with a low MIC value, despite its relatively low content of XH, cohumulone, n+adhumulone, colupulone and n+adlupulone. Thus, we could assume that the main difference between samples may be the remaining hop compounds that are unidentified in this study. Based on this study, we proved different mechanisms of antimicrobial activity against two Gram-positive bacteria with different hop genotypes. Moreover, *Staphylococcus aureus* was more sensitive to the HAE extracts in comparison to *Lactobacillus acidophilus*. An important result of our study is also the observation of giant differences between different hop genotypes in their antimicrobial activity against *Staphylococcus aureus* and *Lactobacillus acidophilus*.

In order to determine which of the quantified components from the extract is the most active, the antimicrobial activity of purified components was also examined. Results are presented in Table 5. It is evident that the β—AF component is the most active against *Staphylococcus aureus*. A weaker antimicrobial activity against *Staphylococcus aureus* for αβ—AF and the lowest antimicrobial activity for XH were determined (Table 5). Similar to our results, MIC values were published for β-acids colupulone (39–78 µg/mL) and lupulone (0.6–1.2 µg/mL) which displayed the greatest antimicrobial activity with the lowest MIC values, while MIC values for α-acids were higher (cohumulone 156–313 µg/mL and humulone 78–156 µg/mL) [32]. The main difference between our results and the results from the published investigation [32] is in the MIC value for XH. Our results, presented in Table 5, show that XH has a high MIC value. Moreover, we were not even able to dissolve a sufficient quantity of XH in the media without precipitation to determine the exact MIC value. However, Bocquet et al. [32] determined the MIC value for XH of 9.8–19.5 µg/mL against different strains of *Staphylococcus aureus*. Similar to our antimicrobial testing of purified samples, Cermak et al. [36] tested the α-acids (the mixture of homologues), β-acids (the mixture of homologues) and pure XH against different strains of *Bacteroides fragilis* and *Clostridium perfringens*. Their results exhibited MIC values of α-acids ranging from 160 to 1540 µg/mL for *Bacteroides fragilis* strains and from 680 to 1370 µg/mL for *Clostridium perfringens* strains. In comparison to α-acids, the β-acids exhibited lower MIC values for *Bacteroides fragilis*, that is 50 to 430 µg/mL and 150 to 430 µg/mL for *Clostridium perfringens*. Therefore, our study confirmed that both α- and β-acids are active against *Staphylococcus aureus* with β-acids being more potent. Both α- and β-acids are also more potent in comparison to XH. Next to experimental results, this is also confirmed by the low negative correlation between XH content of HAE extracts and their MIC against *Staphylococcus aureus*. The results for α- and β-acids are in good agreement with the literature values [32], while the MIC value for XH is much higher in our study. This could probably be explained by using different bacterial strains of *Staphylococcus aureus*.

After testing the individual purified extracts, the β-AF was displayed as the most effective against *Lactobacillus acidophilus* (average MIC value of 20.8 µg/mL), similar to tests against *Staphylococcus aureus*. A weaker antimicrobial activity against *Lactobacillus acidophilus* was determined for αβ-AF (average MIC value of 26.1 µg/mL) and for XH with the highest MIC value (>500.0 µg/mL) (Table 4). Likewise, we were able to determine the MBC values (500.0 µg/mL) for αβ-AF and β-AF. The latter result, in combination with the results of HAE extracts for MBC values against *Lactobacillus acidophilus* (MBC values 187.5–750 µg/mL), provide an important insight since *Lactobacillus* spp. exhibited high resistance against antibiotics [9].

Antimicrobial tests revealed that *Staphylococcus aureus* is more susceptible to HAE compared with *Lactobacillus acidophilus*. However, the fact is that hop extracts with lower MIC and MBC values against *Lactobacillus acidophilus* achieve low MIC and MBC values against *Staphylococcus aureus* as well. Therefore, there is a tendency for the same hop compounds to contribute to the antimicrobial activity of hop extracts in our study and these compounds are unidentified in our study.

Chemical compounds showed strong positive correlations (Table 6) between XH and colupulone (R = 0.836), cohumulone and n+adhumulone (R = 0.873), n+adlupulone and colupulone contents (R = 0.867). On the other hand, as expected, a strong inverse correlation was observed between the amount of unidentified compounds from hop extract versus XH (R = −0.829), cohumulone (R = −0.877), n+adhumulone (R = −0.934), colupulone (R = −0.878) and n+adlupulone contents (R = −0.789). Importantly, the minimal inhibitory concentration against *Lactobacillus acidophilus* (MIC La) demonstrated a strong inverse correlation to XH (R = −0.820), cohumulone (R = −0.952), n+ adhumulone (R = −0.965), colupulone (R = −0.827) and n+adlupulone contents (R = −0.723). However, the MIC La positively correlated with the amount of unidentified compounds (R = 0.943), suggesting that the identified chemical compounds are directly responsible for the low values and the observed antimicrobial activity. Similar positive/negative trends in correlation were observed for the minimal bactericidal concentration (MBC La), but the correlations were weaker (R = 0.606). Likewise, correlations between the antimicrobial activity against *Staphylococcus aureus* (MIC St and MBC St) and other chemical compounds showed similar trends as the ones observed for *Lactobacillus acidophilus*, but correlation coefficients were again generally weaker. Not surprisingly, MIC and MBC values all demonstrated a strong to very strong correlation.

### 2.5. The Effect of Growth Medium Strength on MIC

Besides the determination of MIC values for hop extracts, another important goal of our study was to determine the effect of growth medium strength on these values. Different medium strength could demonstrate (in a laboratory scale) suboptimal conditions for bacteria, such as the ones in the food and feed industry, cosmetics or veterinary medicine applications. Namely, antimicrobial testing of different plant extracts is performed in various media, with important effects on MIC values. For this purpose, we prepared different medium strengths of Mueller Hinton Broth (MHB) medium to test their effect on MIC values of hop extracts. The growth medium strength was increased from half to one and a half concentration, recommended by the producer [37]. The effect of growth medium strength on MIC was measured for *Staphylococcus aureus*. Although the media strength (Figure 2) had only a limited effect on the MIC values, it still gives us some insight into the mechanism of antimicrobial action of hop extract components, demonstrating that direct interactions of extract components with the growth media are of minimal importance, especially when compared to tannins [38]. Even though MIC values vary with the medium strength, a clear trend cannot be observed.

Conclusions of several studies indicate the suitability of hop extracts in the food and feed industry, cosmetics and veterinary medicine applications [10,12,16,18,25,39,40]; however, it needs to be considered that in all described situations the bacterial growth conditions are not as favorable as in commercially prepared media. Therefore, the investigation of MIC and MBC values when the conditions of bacterial growth are suboptimal should be performed using analogous procedures, e.g., medium diluting and concentrating. On the other hand, it is important to understand the growth medium strength effect on the MIC values to facilitate a direct comparison of different studies. A previous study [37] revealed an important effect of growth medium strength on MIC values and the authors linked this fact to tannins’ direct interactions with growth medium components. Meanwhile, the MIC values of hop components are similar regardless of the MHB medium concentration used, confirming previous studies where the mechanism of hop component antimicrobial activity was investigated (hop components integrate into the bacterial cell membrane and affect its metabolism) [10,39]. Even though MIC values of hop extracts were similar in different MHB medium strengths, certain fluctuations could still be observed. Therefore, a standardized protocol for the MIC determination should be established as it has already been for antibiotics.

### 2.6. Determination of Generation Time and Lag Time Duration

By using the plotted dependency graphs of OD_595_ on time, the generation time and lag time durations were calculated according to the growth medium strength. With respect to the scarcity of growth kinetic analyses in the determination of bacterial susceptibility against natural compounds [41], the associated parameters were calculated using Equation (1). For this reason, three samples (HAE Styrian Eagle, β-AF and αβ-AF) were included to investigate the growth of *Staphylococcus aureus*. Increasing concentrations of hop extracts (concentrations lower than the MIC) are generally reflected in a prolonged lag phase duration. The lag phase represents the first phase of the microbial growth curve, following inoculation and preceding the growth of the bacterial population at the end of the lag phase. The delay in the growth during the lag phase appears because of the bacterial adaptation to new circumstances, in order to begin to exploit new environmental conditions [42,43]. Limited reports are available to understand the antibiotic effects on the lag phase, due to low metabolic rates of cells and not enough bacterial material for analyses at this stage [41,42]. It has been assumed that the lag phase is involved in the adaptation of bacteria to conditions in the new media and may be influenced by several factors such as inoculum volume, physiological cells history and the characteristics of the original and the new growth medium [38,42,43]. It has been already reported that an increase in the concentration of the antimicrobial agent results in the extension of the lag phase [38,41,42,43]. As expected, the described studies comprise the greatest extent of antibiotics. Some reports of an extended lag phase and reduced growth by different tannins are available [38,44], on the contrary no studies of hop extracts or hop components as antimicrobial agents include a microbial growth kinetic analysis. The extension of the lag phase for *Staphylococcus aureus* was observed with the increasing concentration of HAE Styrian Eagle or αβ—AF extracts at all media strengths (Appendix A). On the other hand, increasing concentration of β—AF extract exhibited the extension of the lag phase only at 75% and 150% media strengths (Figure 3). Bacteria use the lag phase extension as one of the mechanisms to adapt under stress conditions caused by antimicrobial agents. Bacteria synthesize enzymes and uptake the essential nutrients that are crucial for the cell growth and division [38]. On the other hand, hop resins (α-acids and β-acids) serve as mobile-carrier ionophores that catalyze the cell processes after they integrate into the bacterial cell membrane [10]. The electroneutral influx of molecules that are undissociated takes place and leads to proton exchange for divalent cations and the efflux of the formed complex is performed. Consequently, the protons accumulate into the cell and a decreased nutrient uptake leads to cell death [10,39]. This mechanism of inhibition of bacterial growth by hop extracts is further supported by the fact that lupulones and humulones represent nonpolar molecules and consequently exhibit greater hydrophobicity [25].

The average generation time of *Staphylococcus aureus* is 35 min, obtained under aerobic conditions [45]. In accordance with the scientific literature, the average generation time of *Staphylococcus aureus* negative control (considering negative controls of all medium strengths) in this study is 32.9 min. Generation times of bacteria mostly depend on the organism itself and on the incubation conditions. The generation times are generally extended with the increasing hop extract concentration (at concentrations lower than MIC), although the results are not as conclusive as they are for the lag phase duration (Figure 4). This could be a consequence of a diauxic growth of *Staphylococcus aureus* bacteria upon the addition of hop extracts, as well as a consequence of other inequalities in the bacteria growth rate. That makes the generation time rather difficult to determine. Collectively, according to the results obtained in this study, it may be safely assumed that *Staphylococcus aureus* prolongs its generation time to survive under increasing concentrations of hop components.

## 3. Materials and Methods

### 3.1. Plant Materials

Hop cones were obtained from the hop gene bank maintained at the Slovenian Institute of Hop Research and Brewing, Žalec, Slovenia (Savinjski golding, Aurora, Styrian Wolf, Styrian Eureka, Styrian Dragon, Styrian Eagle and Styrian Fox, all Slovenian varieties) as well as at the Hop Research Institute, Žatec, Czech Republic (P169 Canada (a wild Canadian hop), P157 Belgium (a wild Belgium hop), Nugget (a USA variety), Decorativny (a Russian variety), Chocotsu No. 17 (a Japanese variety) and Early Promise (a English variety). Samples of hop cones were picked up from 5 different plants (from the upper, medium and lower part of the plant) at the time of technological ripeness of each included genotype.

The hop cones were air dried up to 10.1–12.1% of final moisture. Samples were vacuum packed and stored in the dark at 4–8 °C.

### 3.2. Hop Extract Preparation

The HAE of each hop genotype were obtained by using 150 mL acetone/water (9:1, v/v). The solvent was added to 10 g of grounded hop cones. The maceration took place overnight (24 h) by stirring in the dark. The extraction mixtures were filtered, the filtrate was collected and the hydroacetonic solvent was evaporated by rotavapor (Buchi, Switzerland, Uster). In the following step, the liquid/liquid extraction of the hop components was performed by adding chloroform/water (1:1, v/v) solvent. The chloroform phase was collected, and chloroform was again removed by rotary evaporation. The obtained hop extracts are referred to as HAE throughout the text. HAE, obtained by this procedure were used for subsequent antimicrobial activity tests.

### 3.3. Purification of Hop Extracts

The β-acids rich fraction (β-AF), and the combination of α-acids and β-acids rich fraction (αβ-AF) was obtained from Hopsteiner (Mainburg, Germany) and from Labor Veritas (Zϋrich, Switzerland), respectively.

Xantohumol (Figure 5) was isolated from the HAE extract of the Styrian Eagle variety by using preparative HPLC and three successive chromatographic steps on reverse phase (C18) chromatographic columns. All preparative chromatographic separations were performed on the puriFlash^®^ 5.250 chromatographic system (Interchim SA, France), equipped with UV/Vis and ELSD detectors and an autosampler. In all cases, three solvents were used (A: water with 0.1% formic acid; B: acetonitrile with 0.1% formic acid; C: isopropanol with 0.1% formic acid). The initial purification was performed on the PF-50C18HP-F0025 flash chromatographic column, using the following gradient: t = 0–20 min, A = 90%, B = 10%, C = 0%; t = 21 min, A = 0%, B = 100%, C = 0%; t = 22–30 min, A = 0%, B = 100%, C = 0%; t = 31 min, A = 0%, B = 0%, C = 100%; t = 32–43 min, A = 0%, B = 0%, C = 100%; t = 44 min, A = 90%, B = 10%, C = 0%; t = 45–48 min, A = 90%, B = 10%, C = 0%. The flash purification was followed with two successive separations on a US10C18HQ-250/212 chromatographic column, using the following gradient: 0–30 min: A = 40%, B = 60%, C = 0%; t = 31 min, A = 0%, B = 0%, C = 100%; 31–38 min, A = 0%, B = 0%, C = 100%; t = 39 min, A = 40%, B = 60%, C = 0%; 39–44 min, A = 40%, B = 60%, C = 0%. After each separation, fractions were analyzed using the HPLC method described below (Section 3.4). Fractions with sufficient purity were collected.

The LC-MS analysis of the sample was performed using Acquity H-Class UHPLC instrument (Waters, USA), equipped with the Acquity TUV detector (Waters, Milford, MA, USA) and Quattro Premier triple quadrupole mass detector (Waters, Milford, MA, USA) calibrated to a mass resolution of 1 Da (0.75 Da at 0.5 peak height). The mass detector was equipped with an atmospheric pressure ionization interface, enabling the ESI or APCI modes of ionization. The MS Scan and Daughter scan modes of operation were employed during data acquisition yielding a “full picture” of the sample components and corresponding structural information about them, respectively.

The separation was performed on Acquity BEH C18 (100 × 2.1 mm, 1.7 µm) column (Waters, Milford, MA, USA). The eluent flow rate was set to 0.25 mL/min. The elution profile used two solvents, 0.1% aqueous formic acid (A) and acetonitrile (B): 0–0.5 min, 5% B in A; 0.5–12.0 min, 5–90% B in A (linear gradient); 12.0–15.0 min, 90% A in B.

The obtained compound’s identity was verified by comparing spectra and retention times to spectra and retention times of a corresponding standard (XH, Sigma, ≥96%—HPLC). Additional identity confirmation of the compound was obtained using 1H NMR and LC/MS spectroscopy [46] and the description of the 1H NMR spectra is included in Appendix A.

### 3.4. HPLC Determination of Chemical Composition of Purified Hop Extracts (HAE)

According to Analytica—EBC 7.7 method [47] HPLC was employed to determine the α- and β-acids in hop extracts. Extracts were filtered through a disposable syringe filter, Chromafil Xtra PET-45/25 (Macherey-Nagel, Dueren, Germany) and a 10 µL injection loop on the HPLC injector was used. The separation was achieved on the Nucleodur 5–100 C18, 125 × 4 mm HPLC analysis column (Macherey-Nagel, Dueren, Germany). The isocratic mobile phase consisting of distilled water, methanol (J.T.Baker, USA) and 85% aqueous solution of ortophosphoric acid (MERCK, Germany, Taufkirchen) in a ratio of 775/210/9 (v/v/v) was used, and the detection was carried out with a Diode array detector (DAD) set at 314 nm for α-and β-acids and 370 nm for detection of xantohumole, respectively. The quantification was performed by the external standard ICE4 (NATECO2, Wolnzach, Mainburg, Germany) for α-and β-acids and by XH 90% (Steiner Hopfen GmBH, Germany). All solvents were of analytical grade purity.

### 3.5. Minimum Inhibitory Concentration (MIC) and Minimum Bactericidal Concentration (MBC) Determination

Broth dilution method was used to determine the antimicrobial susceptibility [48]. In vitro antibacterial activity of hop extracts (HAE, β-AF, αβ-AF) and pure XH was assayed against *Staphylococcus aureus* ATTC 29213 and *Lactobacillus acidophilus* ATTC 4356.

**Sample preparation**. Tested samples (HAE of different hop varieties, β-AF, αβ-AF and pure XH) were dissolved in 100% dimethyl sulfoxide (DMSO, Sigma-Aldrich, St. Louis, MO, USA) before the analysis. High concentrated DMSO hop extract solutions were mixed with microbial media (Table 7) to reach the optimal extract concentration, although the final DMSO concentration (on 96-well microplates) did not exceed 5% (i.e., the concentration without the effect on the microbial growth).

**Preparation of inoculum—*Staphylococcus aureus***. Bacterial culture was prepared in Mueller Hinton Broth (MHB, Sigma-Aldrich, St. Louis, MO, USA). In order to obtain an equal bacterial concentration, the turbidity of the bacterial medium was measured using Tecan Infinite 1000PRO. Moreover, a calibration curve was prepared. A bacterial concentration of 5∙10^5^ CFU/mL [37] was applied in the assay (the final concentration on 96-well plate was 2.5∙10^5^ CFU/mL). An overnight bacterial culture was used for each assay.

**Preparation of inoculum—*Lactobacillus acidophilus***. Bacterial culture was prepared in De Man Rogosa and Sharpe Broth (MRSB, Sigma-Aldrich, St. Louis, MO, USA). To obtain an equal bacterial concentration the turbidity of the bacterial medium was again measured using Tecan Infinite 1000PRO (Switzerland). A bacterial concentration of 5∙10^5^ CFU/mL [37] was applied in the assay (the final concentration on 96-well plate was 2.5∙10^5^ CFU/mL). Two-day (at 35 °C) bacterial culture was used for each assay.

**The antimicrobial activity assessment.** To evaluate the antimicrobial activity of investigated samples, a broth microdilution assay was used on 96-well microplates. Optical density was determined spectrophotometrically by measuring OD_595_ on Tecan Infinite 1000PRO. Prepared samples were placed into 96-well microplates and serial dilutions were performed horizontally according to the plates. A total of 100 µL of hop samples (HAE) diluted in media (with 5% DMSO) were placed into every well. The 100 µL medium was added as a background control (for every sample concentration). Positive controls were prepared from the culture medium and bacterial suspension only (without hop extracts). One more positive control was performed in each assay, a medium (containing 5% DMSO) and bacterial suspension. To determine microbial turbidity, 100 µL of inoculum was added into wells [28]. Measurements were performed every 10 min and plate shaking was applied before every measurement [48]. The temperature was set according to the optimal incubation temperature for the corresponding bacteria (Table 7) and the measurements were performed for 24 h [32]. Dependency graphs of OD_595_ versus time were plotted. MICs were determined as the lowest concentration of samples where no microbial growth could be detected. Based on the metabolic activity [37], OD_595_ values lower than 0.05 were interpreted as the ones without bacterial growth.

Minimum bactericidal concentration (MBC) values for each bacterium were also determined. For this purpose, bacterial medium agar plates were prepared (Table 7). In total, 100 µL of mixture (bacterial suspension with tested sample) was taken from the microtiter plate used for the MIC determination and inoculated onto agar plates. Agar plates were incubated aerobically for 24 h for *Staphylococcus aureus* and for 48 h for *Lactobacillus acidophilus* at the bacterial optimal temperature (Table 6). MBC values were determined as the lowest concentration of the sample that kills ≥ 99.9% of the initial bacterial inoculum.

### 3.6. Statistical Analyses

Mean values and standard deviation were calculated using Excel (Microsoft, New York, NY, USA). Correlations between the chemical composition and the MIC values against *Staphylococcus aureus* and *Lactobacillus acidophilus* were determined using SPSS Statistic (IBM, version 21, Armonk, NY, USA). The Spearman correlation coefficient was applied to interpret the correlations. Here, R < 0.2 indicates a very weak correlation, R = 0.2–0.4 indicates a weak correlation, R = 0.4–0.6 indicates a moderate correlation, R = 0.6–0.8 indicates a strong correlation and R = 0.8–1 indicates a very strong correlation. Likewise, decreasing negative values indicate a stronger inverse correlation.

### 3.7. The Effect of Growth Medium Strength on MIC

The concentration of the MHB medium used was varied. The medium strength was changed from half to one and a half of the concentration recommended by the producer. Medium concentrations of 50%, 75%, 100% and 150% were used. The effect of the growth medium strength was studied for *Staphylococcus aureus* and the samples β-AF, αβ-AF and HAE of Styrian Eagle.

### 3.8. Determination of Generation and Lag times

To obtain the bacterial growth parameters, a fitting of Equation (1) [42,49] to the experimental growth curve was performed.
(1)$$\ln\frac{N}{N_{0}}=A\exp\left\{-\exp\left[\frac{µe}{A}\left(\lambda -t\right)+1\right]\right\}$$
where *N* represents the cell number at time *t* and *N*_0_ is the cell number at the beginning (t = 0). The cell number is determined by OD_595_ measurement. *µ* represents the maximum specific growth rate and A the maximum number of a cell during the experiment. *λ* represents the lag time duration. The fitted parameters were obtained in Excel by using the least square method. The model was fitted to samples that were used in the determination of the growth medium strength effect on MIC. Thus, the sample concentrations used in this model were only those below the MIC value and the positive controls. According to the observed diauxic growth for all included hop samples, a fitting model was optimized until optical density had stabilized. The graph of the generation time and lag time duration against the sample concentration and medium strength was plotted.

## 4. Conclusions

Extracts (hydroacetonic) of diverse hop genotypes (Slovenian and world-wide) exhibit different antimicrobial activities against two Gram-positive bacteria (*Staphylococcus aureus* and *Lactobacillus acidophilus*). Styrian Dragon, Aurora, Styrian Wolf and Dekorativny exhibited the greatest antibacterial activity against *Staphylococcus aureus*, whereas Savinjski golding, Aurora, Styrian Dragon and Chocotsu No.17 exhibited the greatest antibacterial potential against *Lactobacillus acidophilus*. When taking into consideration the chemical composition of hop extracts and their antimicrobial activity, XH contributes less than α-acids and β-acids. The antimicrobial activity comparison of different hop components (XH, αβ-AF extract, β-AF extract) was determined, showing the best results for the β-AF extract (where the main determined compounds were colupulone and n+adlupulone). Strong inverse correlations of MIC and MBC values were obtained with XH, cohumulone, n+adhumulone, colupulone and n+adlupulone contents, suggesting that identified chemical hop compounds are directly responsible for its antimicrobial activity. Moreover, our results comprise a step forward in the mechanistic understanding of how bacteria *Staphylococcus aureus* fight against hop compounds in the environment by inducing lag phase extension and generation time prolongation. We also demonstrated that the concentration of the growth medium does not significantly affect the MIC values; therefore, the antibacterial activity of hop extracts is not a consequence of their direct interactions with the growth medium. Different hop extracts used in our study rank as potential antibacterial agents which could be applied in pharmaceutical, veterinary, food and cosmetic industries.

## Figures and Tables

**Figure 1 plants-12-00120-f001:**
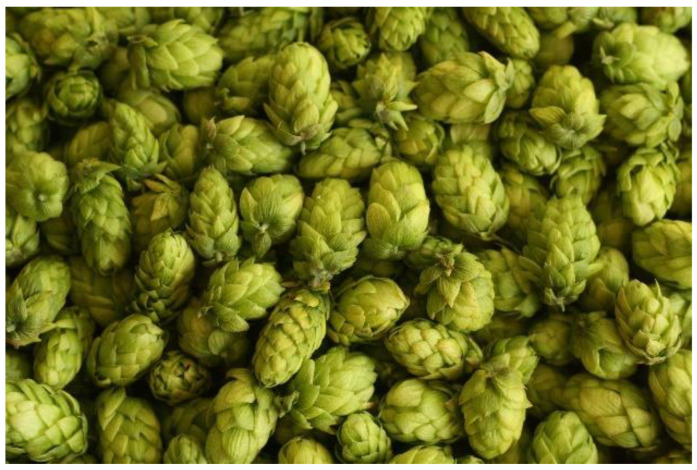
Figure of hop (*Humulus lupulus L*.) cones.

**Figure 2 plants-12-00120-f002:**
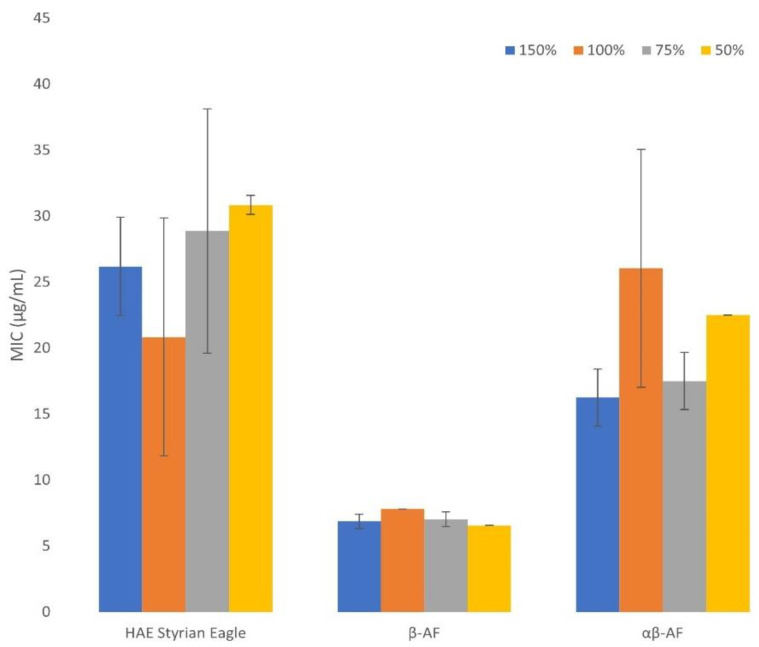
The effect of growth medium strength on MIC values for *Staphylococcus aureus*. HAE Styrian Eagle stands for the hydroacetonic purified extract of Styrian Eagle hop variety, β-AF stands for the β acids rich fraction, αβ-AF stands for the αβ acids rich fraction. The medium strength is described in percentage, the 100% medium strength means it was prepared as recommended by the producer. The 50% medium strength means half of the concentration recommended by the producer. The 75% medium strength represents three quarters of the concentration recommended by the producer. The 150% medium strength represents one and a half of the concentration recommended by the producer. Average values (n = 3) with corresponding standard deviations are presented.

**Figure 3 plants-12-00120-f003:**
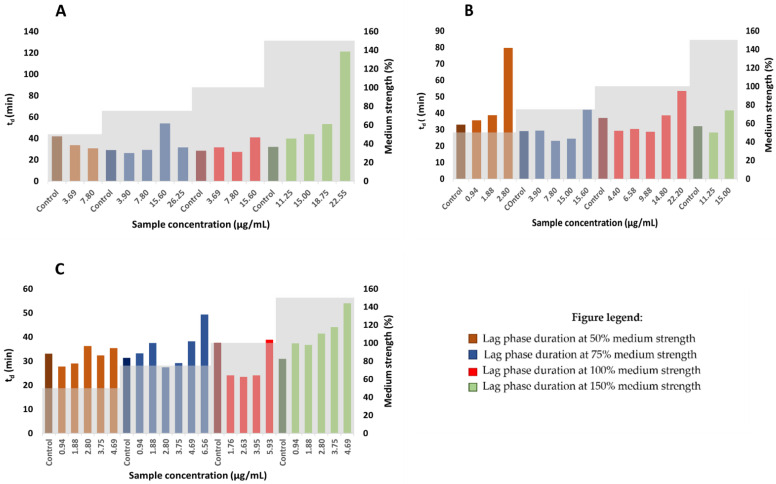
The effect of different concentrations of hop extracts on the lag phase duration of *Staphylococcus aureus*; (**A**) HAE Styrian Eagle, (**B**) αβ-AF extract, (**C**) β-AF extract.

**Figure 4 plants-12-00120-f004:**
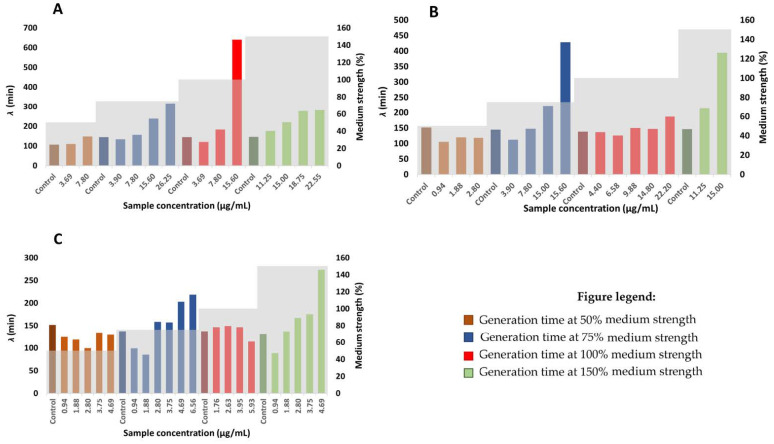
The effect of different concentrations of hop extracts on the generation time of *Staphylococcus aureus*; (**A**) HAE Styrian Eagle, (**B**) αβ-AF extract, (**C**) β-AF extract.

**Figure 5 plants-12-00120-f005:**
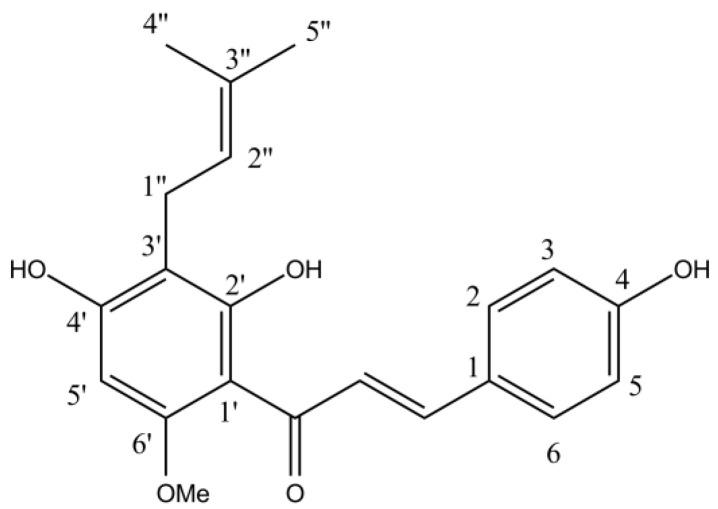
Xanthohumol chemical structure.

**Table 1 plants-12-00120-t001:** The chemical composition of pure hop cones of different hop varieties included in our study, evaluated by HPLC.

Samples	Xanthohumol	Cohumulone	n+adhumulone	Colupulone (%)	n+adlupulone
	Hop Cones (%, w/w)	Hop Cones (%, w/w)	Hop Cones (%, w/w)	Hop Cones (%, w/w)	Hop Cones (%, w/w)
Aurora	1.39	2.83	9.10	2.19	1.91
Savinjski golding	1.39	2.83	9.10	2.19	1.91
Styrian Wolf	0.61	1.10	3.45	1.04	1.13
Styrian Dragon	0.61	1.10	3.45	1.04	1.13
Styrian Eureka	1.19	3.16	13.88	1.72	2.54
Styrian Fox	1.62	2.68	10.37	1.64	1.74
Styrian Eagle	0.70	1.52	5.99	2.65	3.91
Chocotsu No.17 S168 Japan	0.51	0.98	2.33	0.67	0.56
Nugget (USA) S222	0.59	0.26	1.18	1.01	1.73
Belgium S367 P157	0.24	0.25	1.10	0.62	0.85
Dekorativny (Russia) S248	1.54	2.04	6.50	1.52	1.72
Early promise (England) S68	1.55	2.86	10.12	1.97	1.91
Canada P169 S369	4.74	0.50	1.18	1.15	1.43
Caucasus S353 P15	1.03	0.65	2.19	1.49	1.99

**Table 2 plants-12-00120-t002:** The chemical composition of HAE extracts of different hop varieties included in our study, evaluated by HPLC.

Samples	Xanthohumol	Cohumulone	n+adhumulone	Colupulone	n+adlupulone	UI *
	(%, w/w)	(%, w/w)	(%, w/w)	(%, w/w)	(%, w/w)	(%, w/w)
Aurora	2.49	10.95	34.50	8.96	8.08	35.02
Savinjski golding	1.17	4.90	14.41	4.61	5.26	69.65
Styrian Wolf	2.90	14.26	47.38	9.45	9.34	16.66
Styrian Dragon	0.97	4.70	18.50	8.41	13.35	54.07
Styrian Eureka	0.42	1.57	6.17	0.96	1.11	89.76
Styrian Fox	1.65	9.47	25.75	6.13	6.14	50.86
Styrian Eagle	2.00	11.52	47.80	6.58	10.80	21.31
Chocotsu No.17 S168 Japan	2.23	2.78	9.88	7.54	13.57	64.01
Nugget (USA) S222	0.90	2.44	7.51	2.26	2.62	84.26
Belgium S367 P157	1.97	5.14	14.60	6.80	8.22	63.27
Dekorativny (Russia) S248	1.05	1.84	5.76	3.19	4.73	83.42
Early promise (England) S68	0.88	3.47	7.89	2.46	2.37	82.93
Canada P169 S369	0.86	4.17	4.52	3.86	2.52	84.08
Caucasus S353 P15	0.70	1.74	7.07	3.19	4.62	82.68

* UI—unidentified.

**Table 3 plants-12-00120-t003:** The chemical composition of pure components/purified extracts included in our experiment, evaluated by HPLC.

Samples	Xanthohumol (%, w/w)	Cohumulone (%, w/w)	n+adhumulone (%, w/w)	Colupulone (%, w/w)	n+adlupulone (%, w/w)	UI * (%, w/w)
αβ-AF	0.03	12.76	72.79	4.90	4.44	5.07
β-AF	0.00	0.21	0.61	16.90	15.23	67.05
XH	97.99	0.00	0.00	0.00	0.00	2.01

* UI—unidentified.

**Table 4 plants-12-00120-t004:** The MIC and MBC values of investigated hydroacetonic hop extracts (HAE) determined against *Staphylococcus aureus* and *Lactobacillus acidophilus*.

HAE Sample	*Staphylococcus aureus* ATTC 29213		*Lactobacillus acidophilus* ATCC 4356	
	MIC * (µg/mL)	MBC (µg/mL)	MIC * (µg/mL)	MBC (µg/mL)
Aurora	15.6 ± 11.0	31.3	62.5 ± 0.0	375.0
Savinjski golding	19.5 ± 7.8	31.3	62.5 ± 0.0	375.0
Styrian Wolf	15.6 ± 0.0	31.3	83.3 ± 36.1	375.0
Styrian Dragon	9.8 ± 3.9	15.6	62.5 ± 0.0	375.0
Styrian Eureka	19.5 ± 7.8	31.3	104.2 ± 36.1	750.0
Styrian Fox	19.5 ± 7.8	31.3	83.3 ± 36.1	93.8
Styrian Eagle	19.5 ± 7.8	31.3	104.2 ± 36.1	187.5
Chocotsu No.17 S168 Japan	27.3 ± 7.8	62.5	62.5 ± 0.0	375.0
Nugget (USA) S222	31.3 ± 0.0	62.5	125.0 ± 0.0	750.0
Belgium S367 P157	31.3 ± 0.0	125.0	208.3 ± 72.2	750.0
Dekorativny (Russia) S248	15.6 ± 0.0	31.3	104.2 ± 36.1	187.5
Early Promise (England) S68	54.7 ± 15.6	62.5	83.3 ± 36.1	˃750.0
Canada P169 S369	˃250.0	˃250.0	62.5 ± 0.0	750.0
Caucasus S353 P15	˃250.0	˃250.0	83.3 ± 36.1	750.0

* MIC values were determined in four replicates, the average ± standard deviation is given in the table.

**Table 5 plants-12-00120-t005:** The MIC and MBC values of investigated purified samples against *Staphylococcus aureus* and *Lactobacillus acidophilus*.

Purified Sample	*Staphylococcus aureus* ATTC 29213		*Lactobacillus acidophilus* ATCC 4356	
	MIC * (µg/mL)	MBC (µg/mL)	MIC * (µg/mL)	MBC (µg/mL)
αβ-AF	27.3 ± 7.8	62.5	26.1 ± 9.0	500.0
β-AF	7.8 ± 0.0	31.3	20.8 ± 9.0	500.0
XH	˃1250.0	˃1250.0	˃500.0	˃500.0

* MIC values were determined in four replicates, the average ± standard deviation is given in the table.

**Table 6 plants-12-00120-t006:** Spearman’s rank correlation coefficients (R) between chemical parameters and antimicrobial properties of hop extracts.

Parameter	Xan	Coh	Nadh	Col	Nadl	Ui	MIC La	MBC La	MIC St	MBC St
Xan	1.000	0.780 **	0.785 **	0.836 **	0.789 **	−0.829 **	−0.820	−0.601	−0.492	−0.326
Coh	0.780 **	1.000	0.873 **	0.770 **	0.587 *	−0.877 **	−0.952 **	−0.486	−0.385	−0.345
Nadh	0.785 **	0.873 **	1.000	0.755 **	0.719 **	−0.934 **	−0.965 **	−0.537 *	−0.506	−0.505
Col	0.836 **	0.770 **	0.755 **	1.000	0.867 **	−0.878 **	−0.827 **	−0.465	−0.519	−0.337
Nadl	0.789 **	0.587 **	0.719 **	0.867 **	1.000	−0.798 **	−0.723 **	−0.577 **	−0.490	−0.361
Ui	−0.829 **	−0.877 **	−0.934 **	−0.878 **	−0.798 **	1.000	0.943 **	0.606 *	0.494	0.416
MIC La	−0.820 **	−0.952 **	−0.965 **	−0.827 **	−0.723 **	0.943 **	1.000	0.521	0.459	0.425
MBC La	−0.604 *	−0.486	−0.537 *	−0.465	−0.577 *	0.606 *	0.521	1.000	0.681 *	0.678 *
MIC St	−0.492	−0.385	−0.506	−0.519	−0.490	0.494	0.459	0.681 **	1.000	0.933 **
MBC St	−0.326	−0.345	−0.505	−0.337	−0.361	0.416	0.425	0.678 **	0.933 **	1.000

**Abbreviations: Xan**-Xanthohumol; **Coh**-Cohumulone; **Nadh**-n+adhumulone; **Col**-Colupulone; **Nadl**-n+adlupulone; **Ui**-Unidentified; **MIC La**-MIC against *Lactobacillus acidophillus*; **MBC La**-MBC against *Lactobacillus acidophillus*; **MIC St**-MIC against *Staphylococcus aureus;* **MBC St**-MBC against *Staphylococcus aureus*. ** Correlation is significant at the 0.01 level. * Correlation is significant at the 0.05 level.

**Table 7 plants-12-00120-t007:** List of microorganisms and media included in the experiment. The incubation conditions are also reported.

Microorganism	*Staphylococcus aureus* ATTC 29213	*Lactobacillus acidophilus* ATCC 4356
Broth (for MIC)	Mueller Hinton Broth	De Man Rogosa and Sharpe Broth
Agar (for MBC)	Mueller Hinton Agar	De Man Rogosa and Sharpe Agar
Incubation temperature	37 °C	35 °C
Preculturing before the assay	overnight	two days
Antimicrobial activity test	MIC, MBC	MIC, MBC

## Data Availability

All data generated or analyzed during this study are included in this published article.

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
