# Peer review of "Antimicrobial Properties of Different Hop (Humulus lupulus) Genotypes"

_plants, 2022, doi:10.3390/plants12010120_

Round 1

Reviewer 1 Report

The manuscript presents the results of a study conducted to determine the antimicrobial activity of extracts obtained from different hop genotypes. The effects of xanthohumol, the purified β-acid-rich fraction, as well as the α-acid-containing fraction enriched in β-acids were analyzed. It was found that the hop extracts tested were more effective against Staphylococcus aureus than against Lactobacillus acidophilus.

The manuscript is written carelessly and needs to be corrected.

Major remarks

The introduction is very long-winded and vague. It needs some shortening, especially when it comes to vague considerations of its properties (lines 71-75, 77-81) or considerations of the methodology of the research conducted (lines 114-124).

On the other hand, the introduction lacks information on the properties of the Lactobacillus acidophilus strain, justifying the choice of this strain for the study. I found such information only on page 6.

Point 2 - the general comment is that it would be appropriate to present the results of your research first and then move on to the discussion, rather than posting your results as one possible procedure.

Point 2.1 begins with the sentence: "A recent study was published...". This sentence must be removed from here. It can be included in the last paragraph of the introduction, as one of the motivations for undertaking the research presented here.

Another issue regarding point 2.1. The title talks about determining the composition of extracts by HPLC, and the beginning mentions obtaining extracts. Point 2.1 should refer to extraction, described in more detail than now. Did the authors perform the extraction according to the literature (using ethanol and dichloromethane) and compare with their proposed method (acetone and chloroform)? This section needs to be completed. Another point would be 2.2, this section would be about determining the composition of extracts using HPLC.

In my opinion, Table S1 should be included in the main manuscript so that the amounts of each component in pure hop cones and HAE extracts can be compared.

Item 2.2 is incomprehensible, it should definitely be corrected.

Item 3.3 - Description of the 1H NMR spectrum of xanthohumol can be included in Supplementary, it is a known and commercially available compound.

The conclusions are well formulated, with a caveat. Four extracts showed the highest activity against Staphylococcus aureus, not just two; Styrian Wolf and Dekorativny had similar values to Aurora. In the case of Lactobacillus acidophilus, Aurora was similarly not mentioned.

The NMR spectrum description lacks signals from the 1", 2", 4" and 5" hydrogens. Signals lying at 1.68 and 1.59 are poorly described.

Minor remarks

All names of bacterial strains must be written in italics.

NO2 (line 103) - number 2 should be written as a subscript

(CD3)2SO (line 499) - numbers 2 and 3 should be written as a subscript

Hb C=O (line 501) - instead of b should be b (beta)

What is H-0" (line 503), it seems to be a mistake in the description

Reviewer 2 Report

Dear Authors,

The results of the work entitled “Antimicrobial properties of different hop (Humulus lupulus) genotypes” (plants-2087727) are interesting and newsworthy.

The aim of the study was to examine the antimicrobial activity of 14 different hop varieties, of β-acids-rich hop extract, of α- and β-acids-rich hop extract, and of pure xanthohumol, in order to establish which hop components exhibit the strongest antimicrobial effect against Staphylococcus aureus and Lactobacillus acidophilus.

In my opinion, the work fits in the scope of the PLANTS journal and can be published in it after the introduction of the corrections suggested below:

1) Lines 69-71: the sentence “Hop was traditionally also used for medical purposes, especially for the treatment of sleeping disorders (for this purpose, hop is still used nowadays), for activation of gastric functions and as an antibacterial and antifungal agent.” requires citation

2)     All the Latin names of bacteria and plants should be written in Italics e.g. lines 83, 98, 111, 129, 134, etc.

3) Line 150: the abbreviation HAE should be introduced earlier in the manuscript as well as the abbreviation XH – see line 159

4)  Since abbreviations have been introduced (e.g. XH, HAE)  they should be consistently used in the entire manuscript. Please, correct it in the manuscript; the Suggestion applies to all entered abbreviations

5)     Line 305: “… higher MIC in MBC values…” – what do you mean??? Correct it, please

6)     Lines 278-284: The authors obtained very high MIC values of xanthohumol in comparison to the results of  Bocquet et al. Moreover, they state that they  "were not even able to dissolve a sufficient quantity of xanthohumol in the media without precipitation to determine the exact MIC value. However, Bocquet et al. determined the MIC value for xanthohumol of 9.8 – 19.5 μg/mL against different strains of Staphylococcus aureus." Could the authors try to explain what can be the reason for such large differences in MIC values between the current study and the study of Bocquet et al.

7)     Lines 347, 348: the abbreviation MHB has been introduced in line 347 … - correct it …

8)     Literature positions 44, 45, 49 – authors’ names require correction

Best regards,

Reviewer 3 Report

Dear authors,

The manuscript entitled „Antimicrobial properties of different hop (Humulus lupulus) genotypes” Kolenc et al. describes the antimicrobial activity of hop extracts obtained from different hop genotypes against Staphylococcus aureus and Lactobacillus acidophilus.

 After reading the manuscript, I did not notice any significant errors, nor spelling and stylistic errors, English language and style are also fine. The idea of research was very good.

 The authors have done a fairly large amount of work and received a significant number of results. Conclusions adequate to the conducted research.

 Several changes are recommended, and some clarifications are required:

 Please italicize all bacteria names on pages 3 and 4.

Table 4 please use the same number of decimal places.

Round 2

Reviewer 1 Report

Corrections and additions were made to the manuscript in accordance with my recommendations (thank you), however, this was not done quite correctly. This is because the authors did not remove unnecessary elements, which makes the manuscript unsuitable for publication in its current version. Definitely, everything that is unnecessary should be removed.

Minor remarks

The term xanthohumol XH, xanthohumolXH, XHxanthohumol is repeated many times. It should be either xanthohumol or XH. This should be corrected.

Numbering of Tables (Table 12, Supplementary material, Table S11. Table 34, Table45, Table 4 5, Table 56, Table 67) needs to be corrected.

Numbering of the following points needs to be corrected (remove unnecessary digits)

Line 71 - unnecessary space o dot after [10].

Line 147 - unnecessary subtitle "HPLC determination of the chemical composition of purified hydroacetonic hop extracts (HAE)".

Lines 153-155 - sentence "in our study, a hydroacetonic extraction of hop cones was performed, based on . Tthe comparison of the extraction yield was evaluated before the extraction." needs rewording.

Line 157 - Hhydroacetonic, unnecessary H

Line 161 - The investigated, unnecessary T

Line 162 - chloroform: and water, either a colon or word and

Line 162-163 - hydroacetonic hop extracts were obtained (HAE). [1] - should be hydroacetonic hop extracts (HAE) were obtained. What does [1] refer to?

Line 210-211 and 213 - "High 210 Performance Liquid Chromatography (HPLC)" repeated twice. The full name of the technique once is sufficient.

Lines 214-216 - "The remaining 2.01% of the material isn unidentified and does not contain α-acids nor β-acids. The remaininganother purified extractcs were commercially obtained." What does the word isn mean? Word cluster "remaininganother", typo in "extractcs", this needs to be corrected.

The paragraph contained in lines 273-282 was moved to the introduction. Hence it was to be removed.

Line 329 - "and" is unnecessarily italicized

Lines 353-354 - The sentence "Antimicrobial testing tests revealed that Staphylococcus aureus is more susceptible to 353 HAE compared with Lactobacillus acidophilus. of hop extracts in this study exhibits higher 354 MIC in MBC values against..." makes no sense, it should be written correctly

Line 507 - instead of “over night” should be “overnight”

Lines 526-528 - for what purpose the times were repeated twice: t = 21 min, t = 21-30 min, t = 31 min, t = 31-43 min, t = 44 min, t = 44-48 min

Lines 552-556 - the erroneous description of the H NMR spectrum was to be removed from here, instead it remained with the information that it is in Supplementary material. This description should be removed from here.
